# Synthesis of Natural (−)-Antrocin and Its Enantiomer via Stereoselective Aldol Reaction

**DOI:** 10.3390/molecules25040831

**Published:** 2020-02-14

**Authors:** Venkatachalam Angamuthu, Dar-Fu Tai

**Affiliations:** 1Department of Chemistry, Pondicherry University, Puducherry 605014, India; venkatachalam_83@yahoo.co.in; 2Department of Chemistry, National Dong Hwa University, Shoufeng Township, Hualien County 974, Taiwan

**Keywords:** (−)-antrocin, diastereoselective aldol reaction, chiral resolution, natural product synthesis, non-basic Lombardo olefination

## Abstract

The total synthesis of (−)-antrocin and its enantiomer are presented. Antrocin (−)-**1** is an important natural product which acts as an antiproliferative agent in a metastatic breast cancer cell line (IC_50_: 0.6 *μ*M). The key features of this synthesis are: (a) selective anti-addition of trimethylsilyl cyanide (TMSCN) to *α,β*-unsaturated ketone; (b) resolution of (±)**-7** using chiral auxiliary L-dimethyl tartrate through formation of cyclic ketal diastereomers followed by simple column chromatography separation and acid hydrolysis; (c) substrate-controlled stereoselective aldol condensation of (+)-**12** with monomeric formaldehyde and pyridinium chlorochromate (PCC) oxidation for synthesis of essential lactone core in (−)-**14**; and (d) non-basic Lombardo olefination of the carbonyl at the final step to yield (−)-antrocin. In addition, (+)**-9** cyclic ketal diastereomer was converted to (+)-antrocin with similar reaction sequences.

## 1. Introduction

Sesquiterpene lactones belong to an important class of naturally occurring terpenoids that represent a diverse and unique class of natural products. It is mainly constructed by head-to-tail condensation of three isoprene units, bearing subsequent *cis-* or *trans*-fused cyclic lactone rings (Figure 1) [1,2,3]. The most predominant feature of sesquiterpene lactone is the presence of an *α*-methylene-*γ*-lactone ring attached directly to a six or eight-membered carbon ring [1]. Antrocin (−)-**1** is a cyclic sesquiterpene containing a five-membered *γ*-lactone ring with an exocyclic double bond. It was isolated from *Antrodia camphorata,* a valuable medicinal fungus grown in the inner wall of a heartwood of *Cinnamomum kanehirai*—an indigenous tree in Taiwan. The significant medicinal application of this fungus is well-known to the native Taiwanese for several decades, even long before it was biologically categorized under the class of *Antrodia camphorata* in 1990 [4,5,6,7]. Traditional Chinese medicine practitioners prescribed the extract of this fungus as a remedy for food poisoning, vomiting, and many other purposes [8]. In addition to that, it is also widely used to treat liver-associated diseases and tumorigenic diseases [9,10,11,12,13]. In 1995, Chiang and co-workers first reported the isolation of (−)-**1** from *Antrodia camphorata,* which is mainly responsible for the aforesaid medicinal uses [14]. Moreover, the recent studies on cancer cell lines stimulate enormous interest about (−)-**1** [15,16,17,18,19]. It was demonstrated that antrocin shows significantly higher activity against proliferation of metastatic breast cancer cell lines (MDA-MB-231) as compared to the drugs generally used in present days for cancer treatments (e.g., doxorubicin and cisplatin) [15]. (−)-**1** also inhibits colony formation and has proved to be nontoxic to normal cell lines [15,19], which suggests that it could be used as a superior alternative to the available cancer drugs. The difficulties for purification of (–)-**1** from natural fungal resources triggers us to find an alternative in vitro synthetic route for it. Moreover, an extensive study on (−)-**1** and its derivatives is urgently necessary in order to get a better understanding on its biological activities.

Yang et al. [20] reported the synthesis of (±)-antrocin by using Au(III)-catalyzed tandem reaction as a key step. The same group also reported [21] the synthesis of (−)-**1** from naturally occurring carnosic acid as a chiral pool. Although carnosic acid possesses the appropriate stereocenters for the synthesis, however, their higher manufacturing cost and poor availability limit further development. Meanwhile, isolating the key intermediates from ozonolyses and low temperature reactions require sophisticated instrumental facilities, along with skilled synthetic knowledge.

Another synthetic method for (+)-antrocin was reported [22] by Sheng-Han Huang et al. Their tricyclic lactone core ring was constructed by an intramolecular Diels−Alder (IMDA) reaction of the camphanate-containing triene intermediate. However, this approach is associated with a long process which did not provide the required natural product (−)-**1**. Therefore, the synthesis of (−)-**1** via a simple strategy using inexpensive, readily available starting material 6-methoxy-2-tetralone and typical operations was patented [23]. In this paper, we reported the detailed synthesis of natural product (−)-**1** (overall yield 7%) and its enantiomer (+)-**1** using straightforward methods and simple chemicals starting from 3-methyl-2-cyclohexenone.

## 2. Results and Discussion

Retrosynthetic analysis of the antrocin ((−)-**1** and (+)-**1**) is shown in Scheme 1, indicating that the bicyclic ketals (−)-**8** and (+)-**9** are the key intermediates. Further manipulations to the cyano group and *α*-position to the carbonyl group after acetal deprotection could lead either to the target antrocin (−)-**1** or (+)-**1**. The final step is the formation of the exocyclic double bond of **1**. This could be obtained via one carbon homologation of the Lombardo olefination. The segment of tricyclic **13** lactol could be installed appropriately from bicyclic ketal (−)-**8** via deprotection, followed by substrate-controlled aldol addition. The key intermediate that contains quaternary stereogenic bicyclic ketal (−)-**8** and (+)-**9** can be easily accessed from (±)-**7** by diastereotopic separation. Access to bicyclic core (±)-**7** was contemplated from enone **6** through diastereoselctive anti-trimethylsilyl cyanide (TMSCN) addition. Enone **6** could be obtained from 3-methyl cyclohex-2-enone (**2)** by simple alkylation and intramolecular aldol condensation.

The synthesis of (±)-**7** is shown in Scheme 2. Alkylation of commercially available 3-methyl-2-cyclohexenone (**2**) with iodo compound **3** [24,25] gave enone **4** [26]. Michael addition of **4** by CuI/MeLi provided 3,3-dimethylcyclohexanone **5** in excellent yield [27,28,29]. Compound **5** was then subjected to in situ deprotection of acetal, followed by intramolecular aldol condensation, to give α,β-unsaturated bicyclic ketone **6** in 89% yield (two steps).

After having **6** in hand, selective trans-HCN addition with KCN/NH4Cl [30] carried out in DMF under reflux gave excellent yield (88%), but the ratio of cis/trans-isomer (53:47) was not satisfactory. This unfavorable stereochemical outcome turned our attention to find different methods for anti-HCN addition. After complete examination through the study with different reagents, we found diethyl aluminum cyanide is a suitable reagent for anti-addition. This was generated in situ by mixing five equivalents of triethyl aluminum in toluene and 2.3 equivalents of TMSCN in the presence of THF at 0 °C. The reaction mixture was immediately treated with **6** to produce trans-cyano byclic ketone (±)-**7** as a major product (17:83, cis:trans) [30,31,32,33]. When the hexane/THF solvent mixture was used, the selectivity toward the trans-HCN addition diminished. Compound (±)-**7** was confirmed by X-ray crystallography method, and all other analytical data complied with those published [34].

Resolution of (±)-**7** thus synthesized was achieved by utilization of a chiral auxiliary (+)-dimethyl tartrate through ketal formation (Scheme 3). Two commercially available chiral auxiliaries A ((2*S*,3*S*)-1,4-bis(benzyloxyobutane-2,3-diol) [35] and B ((2*R*,3*R*)-dimethyl-2,3-dihydroxysuccinate) were chosen for the resolution of the compound (±)-**7** (Figure 2). Initially, ketone (±)-**7** with A gave the corresponding ketal as a diastereomeric mixture. Unfortunately, these diastereomers were not separated easily by the column chromatographic method with different solvent mixtures. Secondary, (±)-**7** treated with B gave corresponding ketal (−)-**8** and (+)-**9** as a 1:1 diastereomeric mixture separated by column chromatography on silica gel (long column containing 230–400 mesh silica gel, eluted with only DCM as an eluant). The diastereomeric purity of the ketal (−)-**8** was verified by observing the ^1^H NMR peaks at the methines of the ketal rings of the two diastereomers. The structure of (−)-**8** was further confirmed by the X-ray crystallographic method (Figure 3).

The complete synthesis of (−)-**1** is shown in Scheme 4. Ketal (−)-**8** was selected for further reaction. In this event, ketal (−)-**8** was treated with an excess of ethylene glycol, *p*-toluenesulfonic acid under reflux in toluene to give (–)-**10** in good yield (89%). Deprotection and protection proceeded in a single operation through the intermediate (−)-**7a** (optical rotation [α]^25^_D_ = −50.1 (*c =* 1, CHCl_3_)). The reduction of (−)-**10** using DIBAL-H [36,37,38] at 0 °C in toluene, followed by acid hydrolysis with glacial acetic acid/water mixture (4:1), provided keto aldehyde (+)-**12** directly in 87% [35]. Regioselective α-hydroxymethylation of (+)-**12** was achieved by reacting with its lithium enolate generated under low temperature at −78 °C with an ethereal saturated solution of gaseous formaldehyde in THF [39]. Single lactol (+)-**13** was afforded exclusively in moderate yield (48%), and starting material (+)-**12** was recovered (26%) [40]. The stereochemistry at the C9 position in compound (+)-**13** has been confirmed to be R configuration. Nuclear overhauser effect spectroscopy (NOESY) spectral data strongly supports in favor of R configuration, and a clear nuclear overhauser effect (NOE) was observed between the C9 proton and C5 methyl proton (Figure 4) in compound (+)-**13** [41] (see Appendix A). Oxidation of (+)-**13** with silica-supported pyridinium chlorochromate (PCC) gave lactone (−)-**14** in good yield (80%) [20]. In this stage, the structure of compound (−)-**14** was confirmed by an X-ray crystallographic analysis.

Standard and modified Wittig olefination (Ph_3_PCH_2_ and THF, 0 °C and Ph_3_PCH_2_, toluene, and *^t^*^-^BuOH, rt) [42,43] of compound (−)-**14** gave poor conversions (30–38%). However, non-basic Lombardo reagent (Zn, TiCl_4_, and CH_2_Br_2_) led to smooth olefination of ketone (−)-**14** to (−)-**1** in excellent yield (98%) [44]. The spectral data for (−)-**1**, including the [α] (specific rotation), were identical to the natural product isolated from the fungus. Utilizating of Scheme 4 synthetic procedure, (+)-antrocin **1** was synthesized from ketal diastereomer (+)-**9** (See Scheme 5, [α]^20^_D_ = +118 (*c* 1.00, CHCl_3_) (syn).

The mechanism of kinetic-controlled α-hydroxymethylation of (+)-**12** is proposed in Figure 5. The result obtained from the isolated yield of (+)-**13** (**13**, 48% and **12,** 26%) indicates a substrate-controlled addition. Generally, the addition of formaldehyde on to the enolate is reversible. In the point of steric hindrance at low temperature, comparatively, the transition state **13c** is more favorable than **13a** and **13b**. In this case, the enolate reacted with formaldehyde at C1 β-position rather than C3 α-position. The reason, presumably, is the C-C bond formation will proceed via a bicyclic, chelating transition state with lithium metal (Figure 6) bridged between formaldehyde and β-substituted aldehyde. Therefore, the equilibrium shifts more favored to transition state **13a,** which provides compound (+)-**13** lactol via five-membered cyclic transition state **13d**. The transition state **13b** is unfavorable due to the opposite stereocenter in which the formation of the cyclic transition state is difficult. Moreover, the regioselectivity is determined by the increased stability of alkoxide formed by reaction at C1 position due to the possibility of forming a lactol. The formation of lactol or a cyclic chelating transition state is not possible in the case of either aldol addition to the cyano substituent or C3 position. However, this hypothesis was further checked by a controlled experiment; the reaction of **7** under similar reaction conditions (Scheme 6) gave regioselective products **15** and **16** at a 1:2.5 ratio with starting compound **7** (see Appendix A). As already discussed, in the case of (+)-**12**, no such mixture of regioisomers was observed. The regioselectivity is indeed higher with an aldehyde group at β-position than the cyano group. The compound **15** was identified by converting to a compound identical to lactone **14** after DIBAL-H reduction and PCC oxidation (confirmed by comparing with ^1^H and ^13^C NMR). The stereoselectivity at α-position was confirmed by X-ray crystallography studied after oxidation of **13**.

## 3. Conclusions

The total synthesis of natural product (−)-antrocin **1** and its enantiomer (+)-antrocin have been achieved by a simple method (overall yield, 7%) from the commercially available starting material. The crucial tricyclic core (+)-**13** was generated by substrate-controlled *α*-hydroxymethylation leading to the target molecule. The chelating transition structure with lithium metal plays an important role to change the equilibrium in the transition state. The kinetic-controlled mechanisms to compound (+)-**13** have been proposed (Figure 5). The mechanistic path prefers C1 β-position over C3 α-position at low temperature is realized. We are working on computational studies toward the activation energy of regio- and stereoselective products and starting materials. That will be completed in due course. The lactol (+)-**13** and lactone (−)-**14** have structural similarities like antrocin; therefore, more derivatives and further biological investigations could be performed through this synthetic process.

## 4. Experimental

### General Conditions

All reactions involving air- or moisture-sensitive reagents or intermediates were performed under an inert atmosphere of nitrogen in oven-dried glassware. All reagents and solvents were obtained from commercial suppliers and used without further purification, if not mentioned otherwise. THF and diethyl ether were dried over sodium, monitored with benzophenone ketyl radicals, and distilled prior to use. Dichloromethane (DCM) and toluene were dried over CaH_2_ and distilled prior to use. Thin layer chromatography (TLC) was performed using precoated silica gel 60 F254 plates containing a fluorescent indicator; purification by chromatography was done using silica gel (230–400 mesh). ^1^H and ^13^C NMR chemical shifts are reported in parts per million and referenced to the residual solvent. All spectra were obtained at 25 °C. High-resolution mass spectra were recorded on a Bruker MicroTOF spectrometer (Bremen, Germany) by electron spray ionization in positive-ion mode (M+H or M^+^), as indicated. Optical rotations were recorded on a digital polarimeter (Rudolph Research, Flanders, NJ, USA) at 589 nm and reported as follows: [α]^20^_D_ (concentration in g/100 mL and solvent).

**3-methyl-2-(2-(2-methyl-1,3-dioxolan-2-yl)ethyl)-cyclohex-2-enone (4)**. A suspension of NaH (60% suspension in mineral oil, 1.83 g, 49.9 mmol) in DMSO (25 mL) was stirred at room temperature for 15 min. To this suspension was added a solution of 3-methyl-2-cyclohexenone **2** (5 g, 45.3 mmol) in DMSO (25 mL) at 16 °C. The reaction mixture was stirred at room temperature for 30 min, and then, 2-(2-iodoethyl)-2-methyl-1,3-dioxolane (**3**, 12.05 g, 49.9 mmol) was added. The mixture was stirred for 17 h and quenched with saturated (sat.) NH_4_Cl_(aq)_. The aqueous layer was extracted by EtOAc (3 × 70 mL). The combined organic layer was dried over Na_2_SO_4_, filtered, and concentrated. The crude was further purified by column chromatography on silica gel using EtOAc/hexane (15:85, *V:V*) as eluants (R*_f_* 0.46), to obtained **4** as yellow oil (7.11 g, 70%). ^1^H NMR (400 MHz, CDCl_3_) *δ* 3.85 (m, 4H, H-ethylene acetal); 2.27–2.22 (m, 6H); 1.86 (s, 3H, H-7); 1.83 (t, *J* = 7.04 Hz, 2H, H-9); 1.53–1.49 (m, 2H, H-5); 1.26 (s, 3H, H-10); ^13^C NMR (100 MHz, CDCl_3_) *δ* 198.4, 155.3, 135.2, 109.7, 64.5, 37.8, 37.7, 32.7, 23.5, 22.2, 20.9, and 19.9; and HRMS (ESI) *m**/z* calculated (calcd.) for C_13_H_20_O_3_ (M^+^) 224.1412 found 224.1417.

**3,3-dimethyl-2-(2-(2-methyl-1,3-dioxolan-2-yl)ethyl)cyclohexanone (5).** MeLi (36.7 mL, 0.08 mol, (2.21 M in diethyl ether)) was added over a period of 15 min to slurry of CuI (7.74 g, 0.04 mol) in dry diethyl ether at −78 °C. The mixture was stirred for 1 h at −10 °C and then chilled again to −78 °C. Then ketone **4** (4.56 g, 0.02 mol) in dry diethyl ether (20 mL) was added. The mixture was stirred for 30 min at −40 °C and quenched with sat. NH_4_Cl_(aq)_ (40 mL). The resulting mixture was partitioned between water and ethyl acetate. The organic layers were combined, dried over Na_2_SO_4_, filtered, and concentrated under reduced pressure. The resultant crude was further purified by column chromatography on silica gel using EtOAc/hexane (9:91, *V:V*) as eluant (R*_f_* 0.7) to obtain **5** as clear oil (4.43 g, 92%). ^1^H NMR (400 MHz, CDCl_3_) *δ* 3.90–3.89 (m, 4H, H-ethylene acetal); 2.33–2.18 (m, 2H); 2.8 (d, *J* = 10.68 Hz, 1H); 1.92–1.73 (m, 2H); 1.72–1.60 (m, 2H); 1.58 (t, *J* = 7.12, 2H); 1.44–1.33 (m, 2H); 1.29 (s, 3H); 1.01 (s, 3H); 0.75 (s, 3H); ^13^C NMR (100 MHz, CDCl_3_) *δ* 213.5, 110.0, 64.6, 64.5, 60.9, 41.04, 39.7, 38.8, 37.6, 29.4, 23.6, 23.1, 22.4, and 18.7; and HRMS (ESI) *m**/z* calcd. for C_14_H_24_O_3_ (M^+^) 240.1725 found 240.1734.

**4,4a,5,6,7,8-Hexahydro-5,5-dimethyl-2(3*H*)-naphthalenone (6).** 3 N HCl (54.5 mL) was added to a solution of **5** (7.0 g, 0.029 mol) in methanol (150 mL) at room temperature. The mixture was heated under reflux for 19 h, then cooled to room temperature, and methanol was removed under reduced pressure. Resultant reaction mixture was extracted with ethyl acetate (3 × 100 mL). The combined organic layers were washed with NaHCO_3(aq)_ (20 mL). Organic layer was dried over MgSO_4_, filtered, and concentrated under reduced pressure. The crude product was further purified by column chromatography on silica gel using EtOAc/hexane (6:94, *V:V*) as eluants (R*_f_* 0.84) to obtain **6** (4.91 g, 89%) as a yellowish brown oil. ^1^H NMR (400 MHz, CDCl_3_) *δ* 5.89 (t, *J* = 2.04 Hz, 1H); 2.47–2.38 (m, 2H); 2.29–2.23 (m, 1H); 2.21–2.13 (m, 2H); 2.11–2.03 (m, 1H); 1.83–1.74 (m, 1H); 1.70–1.60 (m, 2H); 1.59–1.43 (m, 2H); 1.05 (s, 3H); 0.81 (s, 3H); ^13^C NMR (100 MHz, CDCl_3_) *δ* 200.0, 165.9, 125.9, 53.5, 47.2, 41.5, 36.8, 36.1, 35.78, 29.7, 21.8, 21.8, and 21.3; and HRMS (ESI) *m**/z* calcd. for C_12_H_18_O (M^+^) 178.1358 found 178.1361.

**trans-Octahydro-8a-cyano-5,5-dimethy1-2(1*H*)-naphthalenone ((±)-7).** TMSCN (0.203 mL, 2.05 mmol) in THF (10 mL) was added to a solution of Et_3_Al (1M solution in toluene, 4.9 mL, 4.66 mmol) in THF (5.5 mL) under N_2_ atm at 0 °C. After 20 min of stirring, a solution of **6** (0.166 g, 0.932 mmol) in THF was added at 0 °C over a period of 30 min. The mixture was allowed to warm to room temperature and refluxed for 20 h, then quenched with sat. NaHCO_3(aq)_ (40 mL) and extracted with EtOAc (3 × 20 mL). The combined organic layers were dried over MgSO_4_, filtered, and concentrated. The resultant residue stirred with 10% HClO_4(aq)_ (12 mL) and THF (12 mL) at room temperature for 40 min. Then, the reaction mixture was diluted with water and extracted with EtOAc (3 × 20 mL). The combined organic layers were washed with NaHCO_3(aq)_ (25 mL), dried over MgSO_4_, filtered, and concentrated. The residue was purified by column chromatography on silica gel using EtOAc/hexane (10:90, *V:V*) as eluants (R*_f_* 0.28) to obtain *trans*-cyano decalone (±)**-7** (0.18 g, 89%) as a white crystalline solid, mp 80–82 °C. ^1^H NMR (400 MHz, CDC1_3_) *δ* 2.65–2.58 (m, 2H); 2.36–2.28 (m, 2H); 2.20–2.14 (m, 1H); 2.07 (d, *J* = 13.4 Hz, 1H); 1.95–1.78 (m, 2H); 1.69–1.58 (m, 2H); 1.48–1.25 (m, 3H); 1.07 (s, 3H); 1.02 (s, 3H); ^13^C NMR (100 MHz, CDC1_3_) *δ* 205.7, 121.9, 53.3, 51.1, 41.3, 41.2, 40.8, 38.6, 33.5, 31.84, 24.2, 20.0, and 19.6; and HRMS (ESI) *m**/z* calcd. for C_l3_H_l9_NO (M^+^) 205.1467 found 205.1475.

**cis-Octahydro-8a-cyano-5,5-dimethy1-2(1*H*)-naphthalenone (cis-(±)-7)**. This compound was purified by column chromatography on silica gel eluting with 10% EtOAc-hexane (R*f* 0.44), Mp 52–54 °C. ^1^H NMR (400 MHz, CDCl_3_) *δ* 2.8 (d, *J* = 14.16 Hz, 1H); 2.46 (dd, *J* = 2.2 Hz, *J* = 14.1 Hz); 2.4–2.3 (m, 2H); 2.18–2.11 (m, 1H); 2.04 (dd, *J* = 4.4 Hz, *J* = 11.9 Hz, 1H); 1.88–1.69 (m, 3H); 1.63 (dt, *J* = 14.28 Hz, *J* = 3.64 Hz, 1H); 1.48 (td, *J* = 3.8 Hz, *J* = 13.3 Hz, 1H); 1.36 (s, 3H); 1.33–1.24 (m, 2H); 0.95 (s, 3H); ^13^C NMR (100 MHz, CDCl_3_) *δ* 206.3, 125.0, 51.1, 45.9, 39.9, 39.9, 34.3, 33.5, 31.0, 30.9, 28.9, 24.1, and 19.9.

**Resolution of ((**−**)-8).** Compound (±)-**7** (1 g, 4.8 mmol) was heated to reflux together with dimethyl L-tartrate (4.5 g, 24 mmol) and 92 mg (0.48 mmol) of *p*-toluenesulfonic acid in dry toluene (30 mL) under nitrogen atmosphere for 48 h in a round-bottom flask equipped with a Dean-Stark trap. After 48 h, reaction mixture was allowed to cool to room temperature and then quenched with saturated sodium bicarbonate solution (~30 mL) and extracted with ethyl acetate (3 x 40 mL). The organic layers were combined, dried over MgSO_4_, and concentrated under reduced pressure. Then, the resultant crude mixture (1.7 g) was further purified by chromatography on silica gels using only DCM (100%) as eluant (a long column loaded with 230–400 mesh size). There were obtained 810 mg (45%) of (−)-**8** as a white crystalline solid. R*_f_* 0.68 on TLC (DCM); mp 95 °C; [α]^25^_D_ −44.7 (*c* 1.0, CHCl_3_) and 806 mg of (+)-**9**; R*_f_* 0.62 on TLC (DCM); mp 177–180 °C; and [α]^25^_D_ +20.6 (*c* 1.00, CHCl_3_).

**(4a*S*,8a*S*)-3,4,4a,5,6,7,8,8a-Octahydro-8a-carbonitrile-5,5-dimethyl-trans-naphthalene-2(1*H*)-one-2[((1*S*,2*S*)-di(methoxy-carbonyl))ethylene acetal] ((**−**)-8).**^1^H NMR (400 MHz, CDC1_3_) *δ* 4.89 (d, *J* = 4.9 Hz, 1H); 4.77 (d, *J* = 4.96 Hz, 1H); 3.80 (s, 6H); 2.19 (dd, *J* = 2.8 Hz, *J* = 13.9 Hz, 1H); 2.11 (dq, *J* = 2.9 Hz, 13.0 Hz, 1H); 1.99–1.96 (m, 1H); 1.92–1.81 (m, 2H); 1.78–1.71 (m, 1H); 1.67–1.55 (m, 3H); 1.51 (dd, *J* = 1.52 Hz, *J* = 13.44 Hz, 1H); 1.29–1.16 (m, 2H); 1.07 (s, 3H); 0.97 (dd, *J* = 2.7, *J* = 11.7 Hz, 1H); 0.92 (s, 3H); ^13^C NMR (100 MHz, CDC1_3_) *δ* 170.2, 169.9, 123.2, 112.7, 52.9, 52.8, 52.3, 47.2, 41.3, 39.4, 38.0, 36.8, 33.3, 32.1, 21.9, 19.9, and 19.4; and HRMS (ESI) *m**/z* calcd. for C_l9_H_28_NO_6_ [M + H]^+^ 366.1917 found 366.1906.

**(4a*R*,8a*R*)-3,4,4a,5,6,7,8,8a-Octahydro-8a-carbonitrile-5,5-dimethyl-trans-naphthalene-2(1*H*)-one-2[((1S,2S)-di(methoxy-carbonyl))ethylene acetal] ((+)-9).**^1^H NMR (400 MHz, CDC1_3_) *δ* 4.89 (d, *J* = 5.0 Hz, 1H); 4.84 (d, *J* = 4.92 Hz, 1H); 3.83 (s, 3H); 3.80 (s, 3H); 2.30 (dd, *J* = 2.7 Hz, *J* = 13.7 Hz, 1H); 2.04–1.97 (m, 2H); 1.9 (dt, *J* = 3.2 Hz, *J* = 13.6 Hz, 1H); 1.85–1.82 (m, 1H); 1.72–1.64 (m, 3H); 1.62–1.50 (m, 2H); 1.29–1.18 (m, 2H); 1.07 (s, 3H); 0.99–0.96 (dd, *J* = 2.24 Hz, *J* = 11.44 Hz, 1H); 0.93 (s, 3H); ^13^C NMR (100 MHz, CDC1_3_) *δ* 169.9, 169.8, 122.8, 112.5, 52.9, 52.9, 52.3, 46.8, 41.3, 39.3, 38.0, 37.1, 33.3, 32.1, 21.9, 20.0, and 19.3.

**trans-Octahydro-8a-cyano-5,5-dimethy1-2(1*H*)-naphthalenone ((**−**)-7) and ((+)-7).** A mixture of (−)-**8** (1 g, 2.73 mmol); 12 N HCl (8 mL); and methanol (36 mL) was heated to reflux for 8 h in a round-bottom flask. The reaction mixture was allowed to cool to room temperature, and the organic solvent was evaporated under reduced pressure. The resultant aqueous layer was extracted with ethyl acetate (3 × 30 mL). The combined organic layers were washed with NaHCO_3(aq)_ (30 mL) solution, dried over MgSO_4_, filtered, and concentrated. The crude product was further purified by column chromatography using EtOAc/hexane (10:90, V:V) as eluants to obtained (−)-**7** (532 mg, 95%) as a white solid; mp 88–89 °C and [α]^22^_D_ −50.1 (*c* 1.0, CHCl_3_).

(+)**-7** was prepared from **(+)-9** using the same procedure to obtained a white solid (534 mg, 95.2%); mp 87–90 °C; and [α]^22^_D_ +49.8 (*c* 1.0, CHCl_3_).

**(4a*S*,8a*S*)-3,4,4a,5,6,7,8,8a-Octahydro-8a-carbonitril-5,5-dimethyl-trans-naphthalene-2(1*H*)-one-2-[ethylene acetal] (10).** Compound (−)-**8** (2.1 g, 5.36 mmol) was refluxed with a mixture of *p*-TSA (20 mg) and ethylene glycol (3.6 mL, 53 mmol) in dry toluene (30 mL) under a nitrogen atmosphere for 4 h in a round-bottom flask equipped with a Dean-Stark trap. After this reaction time, the mixture was cooled to room temperature and then quenched with sat. NaHCO_3(aq)_ (~30 mL). The aqueous layer was extracted with EtOAc (3 × 40 mL). The combined organic layers were dried over MgSO_4_, filtered, and concentrated under reduced pressure. The resultant crude product was further purified by column chromatography on silica gels using EtOAc/Hexane (8:92, *V:V*) as eluants (R*_f_* 0.56) to obtain pure (−)-**10** (1.25 g, 89%); mp 56–58 °C and [α]^25^_D_ −44.9 (*c* 1.00, CHCl_3_). ^1^H NMR (400 MHz, CDCl_3_) *δ* 4.09–4.00 (m, 2H); 3.91 (t, *J* = 6.56 Hz, 2H); 2.06 (d, *J* = 13.68 Hz, 1H); 2.00–1.81 (m, 4H); 1.75–1.65 (m, 1H); 1.61–1.48 (m, 4H); 1.29–1.18 (m, 2H); 1.08 (s, 3H); 0.98 (d, *J* = 12.0 Hz, 1H); 0.93 (s, 3H); ^13^C NMR (100 MHz, CDCl_3_) *δ* 123.6, 107.0, 64.8, 64.3, 52.4, 46.4, 41.4, 39.4, 38.0, 36.2, 33.3, 32.1, 22.1, 19.9, and 19.4; and HRMS (ESI) *m**/z* calcd. for C_15_H_23_NO_2_ (M^+^) 249.1729 found 249.1733.

**(4a*S*,8a*S*)-3,4,4a,5,6,7,8,8a-Octahydro-5,5-dimethyl-2-oxo-naphthalene-8a-carbaldehyde ((+)-12).** DIBALH (1 M solution in hexane, 2.2 mL, and 2.12 mmol) was added to a solution of (−)-**10** (443 mg, 1.77 mmol) in toluene (4.8 mL) under N_2_ atm at room temperature. The mixture was stirred at rt for 2 h. Then, 1 N HCl (20 mL) was added to the reaction mixture at 0 °C, diluted with ethyl acetate, and separated for the organic layer. The organic layer was dried over MgSO_4_, filtered, and concentrated under reduced pressure. The resultant crude mixture was stirred with glacial acetic acid/water (4:1) at 60 °C for 20 min and then cooled to 0 °C. Then, quenched with NaHCO_3(aq)_ (30 mL) under slow addition and then extracted with ethyl acetate (10 × 30 mL). The combined organic layers were dried over MgSO_4_, filtered, and concentrated. The resultant crude product was further purified by column chromatography on silica gels using EtOAc/hexane (11:89, *V:V*) as eluants (R*_f_* 0.31) to obtain **12** as a semi-solid (321.9 mg, 87%); [α]^23^_D_ +5.9 (*c* 1.00, CHCl_3_). ^1^H NMR (400 MHz, CDCl_3_) *δ* 9.91 (s, 1H); 2.56 (dp, *J* = 2.5 Hz, *J* = 15.5 Hz, 1H); 2.35–2.26 (m, 1H); 2.21 (dd, *J* = 2.1 Hz, *J* = 14.7 Hz, 1H); 2.06–1.98 (m, 3H); 1.86 (td, *J* = 4.9 Hz, *J* = 13.04 Hz, 1H); 1.81–1.60 (m, 3H); 1.57–1.53 (m, 1H); 1.40–1.23 (m, 2H); 1.00 (s, 3H); 0.737 (s, 3H); ^13^C NMR (100 MHz, CDCl_3_) *δ* 208.3, 205.7, 53.5, 51.8, 51.5, 41.5, 41.0, 35.4, 33.6, 31.4, 22.0, 21.9, and 19.0; and HRMS (ESI) *m**/z* calcd. for C_13_H_21_O_2_ [M+H]^+^ 209.1542 found 209.1531.

**(1*S*,4a*S*,8a*S*)-decahydro-5,5-dimethyl-2-oxo-naphthalene[1,8a-c]furan-1S(3*H*)-ol ((+)-13). Ethereal solution of formaldehyde:** To a 100 mL single-neck round-bottom flask was added paraformaldehyde (7 g) and dried under vacuum at 80 °C for 1 h to remove moisture completely. Then, the round bottom-flask was fitted with septum and flushed with N_2_ gas, heated at 140–160 °C to generate monomeric formaldehyde gas, which was passed, through cannula by means of a stream of N_2_ gas to one of the single-neck round-bottom flask connected with a transfer arm to another flask and contained (40 mL) dried THF with constant stirring at −78 °C for 40 min. The resultant THF solution was used immediately.

Compound (+)**-12** (0.330 g, 1.584 mmol) in THF (3 mL) was added to a stirred solution of LHMDS (3.48 mL, 3.48 mmol) in THF (4 mL) at −78 °C under N_2_ atm and then stirred for 5 h. To the above reaction mixture was transferred the ethereal solution of formaldehyde (40 mL) poured through the connected transfer arm, also at −78 °C. After the addition was completed, the reaction mixture was allowed to stir for 15 min at the same temperature. Then, the reaction was quenched by drop-wise addition of aqueous NH_4_C1 at −78 °C. The reaction mass was stirred at room temperature for 40 min. The reaction mass was extracted with ether (3 × 30 mL). The combined extracts were washed successively with water and brine and dried over MgSO_4_. Filtration and evaporation of the filtrate afforded a clear oil, which was chromatographed on silica gel using EtOAc/CH_2_Cl_2_ (10:90, *V:V*) as eluants to obtain (+)**-13** (181 mg, 48%) as a white semi-solid. (+)-**12** (98 mg, 26%) was recovered from the column. [α]^23^_D_ +71.1 (*c* 1.00, CHCl_3_). ^1^H NMR (400 MHz, CDCl_3_) *δ* 5.43 (d, *J* = 1.48 Hz, 1H); 4.66 (dd, *J* = 1.5 Hz, *J* = 8.52 Hz, 1H); 3.74 (dd, *J* = 7.04 Hz, *J* = 8.4 Hz, 1H); 2.61–2.57 (m, 1H); 2.41 (d, *J* = 1.36 Hz, 1H); 2.38 (m, 2H); 2.18 (d, *J* = 6.72 Hz); 1.93–1.79 (m, 3H); 1.67–1.50 (m, 3H); 1.35–1.34 (m, 1H); 1.23–1.20 (m, 1H); 1.00 (s, 3H); 0.94 (s, 3H); ^13^C NMR (100 MHz, CDCl_3_) *δ* 209.2, 100.3, 62.3, 57.0, 52.6, 50.6, 42.3, 41.3, 37.3, 34.2, 32.2, 23.0, 21.5, and 19.9; and HRMS (ESI) *m**/z* calcd. for C_14_H_22_O_3_ (M^+^) 238.1569 found 238.1575.

**(1*S*,4a*S*,8a*R*)-decahydro-5,5-dimethyl-2-oxo-naphthol[1,8a-c]furan-1(3H)-one ((−)-14)** PCC (290.8 mg, 1.35 mmol) was added at once to a stirred solution of **13** (160 mg, 0.67 mmol) in dried CH_2_Cl_2_ (20 mL) at room temperature under N_2_ atm. The mixture was stirred for 6 h at room temperature, diluted with 60 mL of diethyl ether, and filtered through a celite bed. The filtrate was concentrated, and the resultant crude product was purified by column chromatographed on silica gel using CH_2_Cl_2_ (R*_f_* 0.84) to obtain (−)-**14** (151 mg, 91%) as a white solid. mp 153–154 °C and [α]^23^_D_ −63.4 (*c* 1.00, CHCl_3_). ^1^H NMR (400 MHz, CDCl_3_) *δ* 4.77 (d, *J* = 9.2 Hz, 1H); 4.30 (dd, *J* = 5.68 Hz, *J* = 9.1 Hz, 1H); 2.71–2.66 (m, 1H); 2.49 (d, *J* = 5.6 Hz, 1H); 2.41–2.32 (m, 1H); 2.20–2.21 (m, 1H); 2.07–2.02 (m, 1H); 1.80–1.70 (m, 1H); 1.67–1.52 (m, 4H); 1.46 (td, *J* = 3.16 Hz, *J* = 13.4 Hz, 1H); 1.28 (td, *J* = 3.28 Hz, *J* = 13.8 Hz, 1H); 1.20 (s, 3H); 1.04 (s, 3H); ^13^C NMR (100 MHz, CDCl_3_) *δ* 207.9, 176.0, 65.2, 59.3, 49.6, 49.1, 42.1, 41.4, 35.7, 33.4, 33.3, 22.1, 22.1, and 18.6; and HRMS (ESI) *m**/z* calcd. for C_14_H_20_O_3_ (M^+^) 236.1412 found 236.1415.

**(−)-Antrocin; (1*S*,4a*S*,8a*R*)-decahydro-5,5-dimethyl-2-methylenenaphthol[1,8a-c] furan-l(3*H*)-one ((−)-1).** Zinc dust (571 mg, 8.73 mmol) was briefly flame-dried under high vacuum in a 20 mL vial fitted with a septum cap. Upon cooling, the vial was flushed with N_2_, and anhydrous THF was added (7.1 mL), followed by 1,2-dibromoethane (82 mg) and TMSCl (9.5 mg, 0.087 mmol). After stirring 10 min, dibromoethane (494 mg, 2.84 mmol) was added. The mixture was cooled to −40 °C, and TiCl_4_ (385 mg, 2.03 mol) was added slowly drop-wise over 5–10 min. The thick, dark gray mixture was warmed to 0 °C over a period of 45 min and then stirred at that temperature for 2 days. After 20 h, a further 1 mL of anhydrous THF was added to assist with stirring. An aliquot of the Lombardo reagent thus-obtained (400 µL, 0.161 mmol, 8 eq) was added to a mixture of lactone (−)-**14** (10 mg, 0.0423 mmol, dried 13 h with high vacuum) in CH_2_Cl_2_ (600 µL) at 0 °C. The mixture was warmed to room temperature and stirred for 30 min, then diluted with Et_2_O (2 mL) and quenched with sat. NaHCO_3(aq)_ (2 mL). The resulting biphasic mixture was stirred vigorously for 5 min, and then the layers were separated and the aqueous layer extracted with Et_2_O (2 × 2 mL). Combined organic layers were washed with brine (2 mL), dried over Na_2_SO_4_, filtered, and concentrated under reduced pressure. The resultant crude product was further purified by column chromatography on silica gel using EtOAc/hexane (10:90) as an eluant (R*_f_* 0.65) to obtain (−)-**1** (9.7 mg, 98%) as a white crystalline solid. mp 97–98 °C and [α]^20^_D_ −118 (*c* 1.00, CHCl_3_). ^1^H NMR (400 MHz, CDCl_3_) *δ* 4.81 (d, *J* = 13.2 Hz, 1H, methylene-H_1_); 4.47 (dd, *J* = 6.8, 9.44 Hz, 1H, methylene-H_2_); 4.14 (dd, *J* = 1.32, 9.44 Hz, 1H, H-10); 2.66 (d, *J* = 6.68 Hz, 1H, H-10); 2.38–2.31 (m, 1H, H-1); 2.27–2.20 (m, 1H, H-3); 2.14 (dq, *J* = 1.72, 10.32 Hz, 1H, H-4a); 1.85–1.73 (m, 2H, H-8); 1.57–1.50 (m, 2H, H-7); 1.49–1.43 (m, 2H, H-6 and H-7); 1.25–1.20 (m, 1H, H-6); 1.17 (s, 3H); 0.93 (s, 3H); ^13^C NMR (100 MHz, CDCl_3_) *δ* 178.3, 146.6, 111.1, 69.3, 54.1, 48.4, 46.5, 41.9, 36.7, 33.2, 33.0, 30.3, 22.3, 22.1, and 18.7; and HRMS (ESI) *m**/z* calcd. for C_15_H_22_O_2_(M^+^) 234.1620 found 234.1624.

**(4aR,5S,8aS)-5-(hydroxymethyl)-1,1-dimethyl-6-oxooctahydronaphthalene-4a(2H)-carbonitrile (15). Ethereal solution of formaldehyde:** To a 100 mL single-neck round-bottom flask was added paraformaldehyde (7 g) and dried under vacuum at 80 °C for 1 h to remove moisture completely. Then, the round-bottom flask was fitted with septum and flushed with N_2_ gas, heated at 140–160 °C to generate monomeric formaldehyde gas, which was passed, through cannula by mean of a stream of N_2_ gas to one of the single-neck round-bottom flasks connected with a transfer arm to another flask containing (40 mL) dried THF with constant stirring at −78 °C for 40 min. The resultant THF solution was used immediately.

Compound (±)-**7** (0.4 g, 1.58 mmol) in THF (4 mL) was added to a stirred solution of LHMDS (3.48 mL, 3.48 mmol) in THF (4 mL) at −78 °C under N_2_ atm and then stirred for 5 h. To the above reaction mixture was transferred the ethereal solution of formaldehyde (40 mL) poured through the connected transfer arm, also at −78 °C. After the addition was completed, the reaction mixture was allowed to stir for 15 min at the same temperature. Then, the reaction was quenched by drop-wise addition of aqueous NH_4_C1 at −78 °C, and the reaction mass was stirred at room temperature for 40 min. Then, the reaction mass was extracted with ether (3 × 30 mL). The combined extracts were washed successively with water and brine and dried over MgSO_4_. Filtration and evaporation of the filtrate afforded a clear oil as a mixture of **15** and **16**, which was separated by column chromatography on silica gel using EtOAc/CH_2_Cl_2_ (10:90, *V:V*) as eluants to obtain **15** (181 mg, 48%) as a white semi-solid. 98 mg of **7** (26%) was recovered from the column.

**Compound 15**:^1^H NMR (400 MHz, CDCl_3_) *δ* 4.12-4.08 (m, 1H); 3.81-3.75 (m, 1H); 2.69 (dd, *J* = 8.0 Hz, *J* = 12.0 Hz, 1H); 2.65–2.59 (m, 1H); 2.46 (dd, *J* = 4 Hz, *J* = 12 Hz, 1H); 2.42-2.33 (m, 1H); 2.28-2.18 (m, 1H); 1.98-1.83 (m, 1H); 1.81–1.67 (m, 2H); 1.58–1.55 (m, 2H); 1.41 (dt, *J* = 4.0 Hz, *J* = 16.0 Hz, 1H); 1.28 (dt, *J* = 4.0 Hz, *J* = 16 Hz, 1H); 1.06 (s, 3H); 1.02 (s, 3H); ^13^C NMR (100 MHz, CDCl_3_) *δ* 209.3, 120.3, 60.2, 58.2, 51.8, 43.0, 41.5, 40.7, 35.8, 33.9, 32.1, 24.7, 20.2, and 19.4.

**Compound 16**: ^1^H NMR (400 MHz, CDCl_3_) *δ* 3.86-3.81 (m, 1H); 3.70-3.67 (m, 1H); 2.64 (d, *J* = 14 Hz, 1H); 2.50–2.47 (m, 1H); 2.36 (d, *J* = 14 Hz, 1H); 2.15-2.06 (m, 2H); 1.83-1.76 (m, 4H); 1.44-1.29 (m, 4H); 1.03 (s, 3H); and 0.99 (s, 3H).

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
