# Peer review of "Synthesis of Natural (−)-Antrocin and Its Enantiomer via Stereoselective Aldol Reaction"

_molecules, 2020, doi:10.3390/molecules25040831_

Round 1
Reviewer 1 Report
The manuscript is so full of errors to result really hard to follow. More annoying of all it's the "distreomer" word, badly reported all through the text. Authors must read and correct all errors and rephrase many sentences. Find some highlighted corrections in the attachment.

Author Response
Dear reviewer,
We are thankful for the valuable and insightful comments. These comments have helped us to improve our manuscript. We have fixed the typo and language problems based on your reviews.
Please find an attachment in which we have corrected, all your suggestions, typo and language errors. We marked as yellow and red color shadows to indicate the changes.
Thank you

Reviewer 2 Report
This manuscript reports the total synthesis of both enantiomers of antrocin, a natural compound showing an interesting anticancer activity. Previous total synthesis of this sesquiterpene have been reported recently (see ref 21-22). The synthetic strategy used here is however original and has the advantage of starting from low cost starting material.
The interest of this synthesis is, in my opinion, sufficient to deserve publication in Molecules after some corrections.
Remarks:
- The text contains many typos and some gramatrical errors. For instance (non-exhaustive):
- pg3, second to last line: "we were tried"
- pg 4, line 115: "gerarated"
- pg 5, line 134: "Although, this case the enolate"
- pg5, line 136: "trasition"
- pg 7, line 159: "ompound"
- pg 7, line 161: "have"
- pg 7, line 162: "impartant", "roll"
- pg 7, line 163: "machanistic", "strically"
- pg 7, line 164: "we have working"
- On page 5, the authors mention transition states II, III and IV but it is explained nowhere to which transition states these later correspond.
- The authors rationalize the observed (C1) regioselectivity for the aldol reaction by formation of a chelating transition structure involving the aldehyde group (see Figure 6). This hypothesis is supported by the change in regioselectivity obtained when replacing the aldehyde by a cyano group (see Scheme 7). Another possible explanation could be that addition onto formaldehyde is in fact reversible and regioselectivity is determined by the increased stability of alkoxide formed by reaction at C1 position due to the possibility of forming a hemiacetal (not possible in the case of C3 reaction or with a cyano group instead of the aldehyde). The authors should consider this latter hypothesis.
Author Response
Answers
Dear Reviewer,
We are thankful for the valuable and insightful comments. Your comments helped us a lot to improve our work. We have fixed the typo and language errors based on your suggestions.
- pg3, second to last line: "we were tried"- corrected and changed to different sentence please check the revised manuscript. - pg 4, line 115: "gerarated"—corrected to “generated”- pg 5, line 134: "Although, this case the enolate"—corrected and changed the different sentences.
- pg5, line 136: "trasition"—corrected
- pg 7, line 159: "ompound"---corrected to compound
- pg 7, line 161: "have"—corrected
- pg 7, line 162: "impartant", "roll"—corrected
- pg 7, line 163: "machanistic", "strically"—corrected the mistakes
- pg 7, line 164: "we have working"—corrected in the revised manuscript.
- On page 5, the authors mention transition states II, III and IV but it is explained nowhere to which transition states have these later corresponded.
--corrected in the revised manuscript
The authors rationalize the observed (C1) regioselectivity for the aldol reaction by formation of a chelating transition structure involving the aldehyde group (see Figure 6). This hypothesis is supported by the change in regioselectivity obtained when replacing the aldehyde by a cyano group (see Scheme 7). Another possible explanation could be that addition onto formaldehyde is in fact reversible and regioselectivity is determined by the increased stability of alkoxide formed by reaction at C1 position due to the possibility of forming a hemiacetal (not possible in the case of C3 reaction or with a cyano group instead of the aldehyde). The authors should consider this latter hypothesis.
Answer: This is a valuable suggestion. We included this hypothesis in the revised manuscript. Thank you very much for your valuable suggestion.
Reviewer 3 Report
In my opinion the article entitled “Synthesis of natural (-)-antrocin and its enantiomer via streoselective aldol reaction” is well written and should be published after some minor corrections. For example in line 10 is impartant. Compound 8 is not a ketol – line 109 and 110.
Author Response
Response to Reviewer 3 Comments
Dear Reviewer,
We are thankful for the valuable and insightful comments. We have fixed the typo and language errors based on your suggestions.
Remarks
- in line 10 is impartant.--corrected in the revised manuscript
-Compound 8 is not a ketol – line 109 and 110.---thank you for your comments, we have been corrected in the revised manuscript.
Round 2
Reviewer 1 Report
The reported synthetic procedure describe the preparation of two enantiomeric natural products. The strategy and the discussion is interesting. Although many typing and style corrections have been made, many errors are still present:
What does it mean "Trimane-type" (line 27)? Other typing/formal errors:
37- significantlyt
48-The same group
49- possesses
57-material
58-detailed
59-straightforward
71-Enone 6 could be obtained…
81-Delete the word “which”
82-satisfactory
83-reagents
89-“Were complies”
95-resolute
97-able to separate
and so on.
Author Response
Response to Reviewer 1 Comments
Dear Reviewer,
We are thankful again for your valuable comments and suggestions. Your comments helped us a lot to improve our work. We have fixed the typo and language errors. Newly corrected sentences were marked as a gray color.
Question:
What does it mean "Trimane-type" (line 27)? Other typing/formal errors:
37- significantlyt
48-The same group
49- possesses
57-material
58-detailed
59-straightforward
71-Enone 6 could be obtained…
81-Delete the word “which”
82-satisfactory
83-reagents
89-“Were complies”
95-resolute
97-able to separate
and so on.
Response
What does it mean "Trimane-type" (line 27)Thank you for your comment; we have changed that part of the sentence. Please check in the revised manuscript.
Significantlyt---corrected The same group…corrected Possesses…corrected in the revised manuscript Material…corrected Detailed…changed Straightforward…changed Enone 6 could be obtained….changed Satisfactory….changed Reagents…corrected Were complies”…..Thank you for the comment. This sentence was corrected. Please check in the revised manuscript. Resolute… Thank you for the comment, changed. Please check in the revised manuscript. able to separate... This sentence was changed. Please check in revised manuscript.Additionally, other formal and typing error was corrected and marked as gray color shadows.